# Ethanol Extract of *Maclura tricuspidata* Fruit Protects SH-SY5Y Neuroblastoma Cells against H_2_O_2_-Induced Oxidative Damage via Inhibiting MAPK and NF-κB Signaling

**DOI:** 10.3390/ijms22136946

**Published:** 2021-06-28

**Authors:** Weishun Tian, Suyoung Heo, Dae-Woon Kim, In-Shik Kim, Dongchoon Ahn, Hyun-Jin Tae, Myung-Kon Kim, Byung-Yong Park

**Affiliations:** 1Bio-Safety Research Institute and College of Veterinary Medicine, Jeonbuk National University, Iksan 54596, Korea; tianws0502@126.com (W.T.); syheo@jbnu.ac.kr (S.H.); iskim@jbnu.ac.kr (I.-S.K.); ahndc@jbnu.ac.kr (D.A.); hjtae@jbnu.ac.kr (H.-J.T.); 2Department of Food Science and Technology, Jeonbuk National University, Jeonju 54896, Korea; eodns3344@gmail.com

**Keywords:** *Maclura tricuspidata* (MT) fruit, neuroprotection, hydrogen peroxide, oxidative damage, MAPK, NF-κB

## Abstract

Free radical generation and oxidative stress push forward an immense influence on the pathogenesis of neurodegenerative diseases such as Alzheimer’s disease and Parkinson’s disease. *Maclura tricuspidata* fruit (MT) contains many biologically active substances, including compounds with antioxidant properties. The current study aimed to investigate the neuroprotective effects of MT fruit on hydrogen peroxide (H_2_O_2_)-induced neurotoxicity in SH-SY5Y cells. SH-SY5Y cells were pretreated with MT, and cell damage was induced by H_2_O_2_. First, the chemical composition and free radical scavenging properties of MT were analyzed. MT attenuated oxidative stress-induced damage in cells based on the assessment of cell viability. The H_2_O_2_-induced toxicity caused by ROS production and lactate dehydrogenase (LDH) release was ameliorated by MT pretreatment. MT also promoted an increase in the expression of genes encoding the antioxidant enzymes superoxide dismutase (SOD) and catalase (CAT). MT pretreatment was associated with an increase in the expression of neuronal genes downregulated by H_2_O_2_. Mechanistically, MT dramatically suppressed H_2_O_2_-induced Bcl-2 downregulation, Bax upregulation, apoptotic factor caspase-3 activation, Mitogen-activated protein kinase (MAPK) (JNK, ERK, and p38), and Nuclear factor-κB (NF-κB) activation, thereby preventing H_2_O_2_-induced neurotoxicity. These results indicate that MT has protective effects against H_2_O_2_-induced oxidative damage in SH-SY5Y cells and can be used to prevent and protect against neurodegeneration.

## 1. Introduction

Neurodegenerative diseases, including Parkinson’s disease (PD) and Alzheimer’s disease (AD), are caused mainly by neuronal damage, which is triggered by oxidative stress in many situations [1,2,3,4]. Endogenous ROS such as hydrogen peroxide (H_2_O_2_) has long been considered destructive molecules, which have been implicated in the pathogenesis of neurodegenerative diseases by causing oxidative stress and inducing apoptosis, and then promoting the pathological process of neurodegenerative diseases [5].

ROS can inactivate enzymes, oxidize proteins, damage DNA, cause lipid peroxidation, and denature proteins, disrupting cell function and integrity. They bring about neuronal injury, necrosis, and apoptosis, resulting in neurodegenerative diseases and loss of other functions [6,7]. ROS are mainly generated in mitochondria and accumulate during aging; therefore, limiting oxidative stress can be neuroprotective [8]. Activating the main antioxidant enzymes in the brain is an important method and mechanism to reduce neuronal damage caused by oxidative stress [9]. Antioxidant enzymes such as superoxide dismutase (SOD), catalase (CAT), and peroxiredoxin family proteins scavenge superoxide and H_2_O_2_, thereby decreasing oxidative stress [5,10].

Regulatory proteins in the Bcl-2 family are important apoptosis-related factors and central moderators of cell death in H_2_O_2_-promoted oxidative stress [11]. Bcl-2 is an anti-apoptotic protein and forms a heterodimer with Bax, an apoptotic activator, thereby determining the fortune of cells [12]. Bax and Bcl-2 proteins are used commonly as indicators of cell survival and apoptosis, and their protein expression ratio (Bax/Bcl-2) can be used to assess cell status. Moreover, the cysteine protease caspase-3 is another vital protein involved in apoptosis that can be activated by H_2_O_2_ [13,14]. In summary, Bcl-2 upregulation, or cleaved-caspase-3 and Bax downregulation, can prevent neuronal apoptosis caused by oxidative stress.

Mitogen-activated protein kinase (MAPK) cascades are important signaling pathways that participate in various cellular processes including apoptosis, oxidative stress, proliferation, and stress responses. Previous studies have revealed that MAPK pathways are activated in neuroblastoma cells in response to H_2_O_2_ treatment [15,16]. In mammalian cells, there are three MAPK families: extracellular signal-regulated kinases (ERKs), C-Jun N-terminal kinases (JNKs), and p38 kinases. Abnormal levels of phosphorylated ERK, JNK, and p38 MAPKs have been found in the brains of Alzheimer’s disease (AD) patients [17], suggesting that MAPKs participate in the pathogenesis of the neurodegenerative disease. Thus, understanding the biochemical processes surrounding H_2_O_2_-induced neuronal apoptosis can provide insight into the pathogenesis of neurodegenerative diseases and the effectiveness of treatments and explore alternative medicine for curing neurodegenerative disease.

Natural products contain a variety of active bioactive compounds that are good for health. The active ingredients in many herbal medicines have been verified to have neuroprotective effects by scavenging free radicals, thereby rejuvenating nerve cells from oxidative stress-induced damage [5,18,19]. MT is produced in high quantities in many Asia countries and is used as traditional medicine [20]. MT extracts and their active ingredients including vitamins, amino acids, and bioactive compounds such as rutin [21], anthocyanins [22], chlorogenic acid [23], and polysaccharides [24] have been verified to have antioxidant and neuroprotective effects. However, the underlying mechanism by which MT extracts exert their antioxidant effects is unknown. Our aim in this study was to explore the role of MT in H_2_O_2_-induced neurotoxicity and determine the underlying molecular mechanisms.

## 2. Results

### 2.1. HPLC Analysis of MT Extract

The chemical composition of MT fruit was evaluated using a Waters HPLC system. Peaks of polyphenolic compounds and parishin derivatives were observed (Figure 1B,D). The contents of polyphenolic compounds and parishin derivatives in the MT extract are shown in Table 1.

### 2.2. Radical Scavenging Activity of MT Extract

As shown in Figure 2 and Table 2, MT scavenged DPPH radicals in a concentration-dependent manner and was an active radical scavenger, with an IC_50_ value of 355.821 ± 8.343 µg/mL. MT also showed antioxidant activity in the ABTS assay, with an IC_50_ value of 278.741 ± 1.300 µg/mL.

### 2.3. MT Improved H_2_O_2_-Induced Cell Damage in SH-SY5Y Cells

To investigate the protective effects of MT against H_2_O_2_-induced cytotoxicity in SH-SY5Y cells, cell viability was evaluated using MTT assays. First, the LC_50_ of H_2_O_2_ damage was determined to be 300 µM (Figure 3A). Cells were treated with MT (5–100 µg/mL) for 24 h to assess the cytotoxicity, and cell viability was significantly (*p* < 0.001) altered at the dose of 100 µg/mL (Figure 3B). The antioxidant effects of MT on H_2_O_2_-challenged cell damage were assessed. Cells were treated with non-toxic doses of MT for 2 h before exposure to 300 µM H_2_O_2_. H_2_O_2_ treatment alone significantly decreased cell viability, while pretreatment with MT at 5 to 50 µg/mL dramatically (*p* < 0.05, or *p* < 0.001) attenuated the H_2_O_2_-induced cytotoxicity, especially the dose of 50 µg/mL (Figure 3C). The protective effect of MT (20, 50 µg/mL) was supported by morphological observations. H_2_O_2_-challenged cells showed a shrunken appearance and a decrease in the number of neurites, and this was ameliorated by MT pretreatment (Figure 3D).

### 2.4. MT Attenuated H_2_O_2_-Induced Nuclear Morphological Changes in SH-SY5Y Cells

To investigate nuclear morphological changes after pretreatment of SH-SY5Y cells with MT before exposure to H_2_O_2_, cells were stained with DAPI [25]. Control cells had regular oval-shaped nuclei, whereas H_2_O_2_-treated cells exhibited nuclear condensation (white arrow) and sporadic weaker staining (red arrow). Percentages of condensed nuclei and fragmented nuclei in the MT pretreatment groups were drastically (*p* < 0.05, or *p* < 0.001) decreased relative to that in the group not pretreated with MT, and there were fewer and less notable H_2_O_2_-induced nuclear morphological changes compared to those seen in cells exposed to H_2_O_2_ only (Figure 4).

### 2.5. MT Inhibited H_2_O_2_-Promoted ROS Production and LDH Leakage in SH-SY5Y Cells

ROS levels are indicators of oxidative stress, and excess ROS generation triggers cell apoptosis. LDH release level in media was measured as an index of cytotoxicity after exposure to H_2_O_2_. Compared with untreated control cells, cells challenged with H_2_O_2_ showed a markedly (*p* < 0.001) increased in ROS generation and LDH leakage, indicating that H_2_O_2_ treatment resulted in oxidative stress and cytotoxicity. However, pretreatment with MT significantly (*p* < 0.05, or *p* < 0.001) blocked the H_2_O_2_-challenged accumulation of ROS and release of LDH (Figure 5A,B). These results demonstrated that MT exerted protective effects on cells exposed to H_2_O_2_ by reducing oxidative stress and alleviating cytotoxicity.

### 2.6. MT Increased the Expression of Antioxidant Enzymes in H_2_O_2_-Exposed SH-SY5Y Cells

Antioxidant enzymes, such as SOD and CAT, play important roles in maintaining an appropriate intracellular oxidation status. In this study, gene expression of SOD and CAT was examined, and treatment of cells with H_2_O_2_ resulted in a markedly (*p* < 0.001) decreased in gene expression of SOD and CAT. However, cells pretreated with MT followed by exposure to H_2_O_2_ demonstrated drastically (*p* < 0.05, or *p* < 0.001) higher gene expression of CAT and SOD than cells treated with H_2_O_2_ only (Figure 6A).

### 2.7. MT Elevated the Expression of Neuronal Biomarkers in Cells Exposed To H_2_O_2_

Gene expression of neuronal biomarkers including brain-derived neurotrophic factor (BDNF), aromatic L-amino acid decarboxylase (AADC), and tyrosine hydroxylase (TH) was assessed. Treatment of SH-SY5Y cells with H_2_O_2_ alone decreased the expression of the neuronal biomarkers BDNF, AADC, and TH (*p* < 0.05 or *p* < 0.01), but levels of these genes were similar (*p* < 0.05, *p* < 0.01, or *p* < 0.001) to those in control cells when SH-SY5Y cells were pre-treated with MT before exposure to H_2_O_2_ (Figure 6B).

### 2.8. MT Regulated H_2_O_2_-Induced Apoptosis-Related Proteins Expression

To further explore the mechanism of H_2_O_2_-challenged apoptosis, the expression of anti-apoptotic Bcl-2 and pro-apoptotic Bax was measured. Cells treated with H_2_O_2_ showed notable (*p* < 0.05, *p* < 0.01, or *p* < 0.001) upregulation of Bax expression and downregulation of Bcl-2 expression compared with the untreated group. Pretreatment with MT protected against the above effects of H_2_O_2_, as characterized by increased Bcl-2 expression and decreased Bax expression as well as a decreased Bax/Bcl-2 ratio (Figure 7A).

Caspase-3 plays a key role in neuronal apoptosis [26]. H_2_O_2_ induces the activation of caspase-3, which can enhance the process of apoptosis. As shown in Figure 7B, H_2_O_2_ treatment markedly elevated the expression of caspase-3, whereas pretreatment with MT significantly (*p* < 0.05, or *p* < 0.001) decreased the expression of caspase-3.

### 2.9. MT Attenuates Oxidative Stress via the MAPK and NF-κB Pathway

Previous studies have demonstrated that H_2_O_2_-challenged neurotoxicity can evoke cell death (apoptosis) via activation of MAPK family proteins [15,16]. Cells exposed to H_2_O_2_ alone showed increased (*p* < 0.05 or *p* < 0.01) expression of phosphorylated ERK, JNK, and p38 MAPK compared to control cells. Pretreatment with MT inhibited H_2_O_2_-induced activation of MAPK family proteins (Figure 8A–C). NF-κB p65 is a critical transcription factor that responds to H_2_O_2_-induced neuronal damage [27]. The level of phosphorylated NF-κB increased (*p* < 0.001) in H_2_O_2_-exposed cells. By contrast, MT pretreatment drastically (*p* < 0.001) protected against the H_2_O_2_-induced increase in phosphorylated NF-κB (Figure 8D). Taken together, these results indicate that MT is capable of protecting neuroblastoma cells from H_2_O_2_-induced damage by blocking MAPK and NF-κB signaling.

## 3. Discussion

Oxidative stress occurs as a result of an imbalance between production and accumulation of oxidants, including ROS, and is very toxic to cells and tissue [28,29]. Accumulation of ROS disturbs cellular homeostasis and activates a series of mechanisms that result in cell destruction. Oxidative stress has been implicated in neuronal cell death in several neurodegenerative diseases including Alzheimer’s disease and Parkinson’s disease. There is great interest in drugs that can reduce neurotoxicity induced by oxidative stress to help prevent and treat neurodegenerative disorders. Most natural medicines are rich in antioxidants and have limited side effects. In this case, looking for potential neuroprotective agents from natural products that can reduce neurotoxicity caused by oxidative stress may help prevent and treat neurodegenerative diseases [30]. Our purpose in this study was to explore the protective properties of MT on oxidative stress-induced neurotoxicity in SH-SY5Y cells and elucidate potential underlying mechanisms.

H_2_O_2_ is an oxidizing compound that functions as an oxidative stress inducer by increasing the production of ROS. H_2_O_2_-induced cytotoxicity is a common method used to investigate potential neuroprotective antioxidant effects of natural products [18,27]. Neural cells that are treated to H_2_O_2_ may suffer an apoptotic-like delayed death and necrosis. Exogenous H_2_O_2_ can pass through the cell membrane, thereby directly changing the intracellular homeostasis, which disturbs signal transduction and then causes DNA damage and cell apoptosis [31,32].

Apoptotic cell death is a programmed physiological process that is accompanied by morphological and biochemical alterations, including cell shrinkage, apoptotic body formation, nuclear condensation, activation of caspases, and chromosomal DNA fragmentation [33,34,35]. In the current study, morphology and DAPI staining showed that the morphology of cells and nuclei have changed after the cell was treated by H_2_O_2_ alone, and MT pretreatment alleviated the cytotoxicity induced by H_2_O_2_, which was further confirmed by MTT assay. Moreover, Intracellular ROS generation and LDH release are related to neuronal apoptosis and necrosis [5,36,37]. Overproduction of intracellular ROS can provoke mitochondrial dysfunction and neurodegeneration [38,39]. Fortunately, HPLC analysis revealed that MT contains many polyphenolic compounds and parishin derivatives, including rutin, caffeic acid, quercetin, and gastrodin (Table 1), all of which have strong antioxidant properties [40,41,42]. A previous study also showed that MT exists antioxidant activity [43]. Moreover, DPPH and ABTS assays revealed that MT has the antioxidant capacity and can scavenge free radicals to decrease oxidative stress, which in turn reduces H_2_O_2_-induced oxidative stress [44,45].

Accumulation of intracellular ROS or the destruction of antioxidant defense mechanism triggers physiological disorder and cell damage. Antioxidant enzymes such as SOD, CAT, and glutathione peroxidase (GPx) are active scavengers of superoxide and hydrogen peroxide. Previous studies have shown that the neuroprotective effects of some phytomedicines, such as MT extract, exist radical scavenging activity, enhance antioxidant enzyme activity, and increase the expression of the antioxidant enzymes CAT and SOD [43,46]. Furthermore, the bioactive compounds, such as protocatechuic acid, caffeic acid, rutin, and so on, can elevate the levels of the antioxidant enzyme [47,48,49]. MT can elevate the decrease in the gene expression of antioxidant enzymes induced by H_2_O_2_. Moreover, BDNF, TH, and AADC play critical roles in the neuronal survival, growth, and differentiation of dopaminergic (DArgic) neurons. Degeneration of these neurons in Parkinson’s disease leads to catecholamine depletion and ultimately reduces dopamine levels in the substantia nigra [50]. MT attenuated H_2_O_2_-induced reduction in expression of BDNF, TH, and AADC. Hence, MT can protect human dopaminergic cells against oxidative stress through a variety of mechanisms to defense against the neurodegenerative disorders caused by oxidative stress. In summary, MT existed the ability to protect the SH-SY5Y cell from H_2_O_2_-induced neurotoxicity by enhancing the expression of antioxidant enzymes and neuronal biomarkers.

H_2_O_2_-induced apoptotic cell death in SH-SY5Y neuroblastoma cells is mediated by the intrinsic pathway of mitochondria through the activation of caspase [51]. The anti-apoptotic protein, Bcl-2, and pro-apoptotic, Bax, play important roles in the mitochondrial-related apoptosis pathway [25]. Bcl-2 family proteins govern apoptosis by regulating the permeability of the mitochondrial outer membrane [52]. Moreover, The Bcl-2 family protein mediates the apoptotic process through balancing pro-apoptotic (Bax) and anti-apoptotic (Bcl-2) expression, and their ratio (Bax/Bcl-2) is a functional indicator in regulating the apoptotic cell death [12]. In this study, H_2_O_2_-induced neurotoxicity slightly increased Bax and decreased Bcl-2 protein expression, all of which was reversed by MT pretreatment. Caspase-3, which is activated by H_2_O_2_, and is an apoptotic executor by triggering DNA fragments [18]. MT attenuated the activation of caspase 3 protein expression. In summary, MT can balance Bax and Bcl-2 protein expression and inhibit caspase-3 activation to protect against H_2_O_2_-induced neuronal damage.

MAPK cascades occupy an important role in transduction extracellular signals to cellular responses and modulate vital cellular processes such as proliferation, metabolism, stress responses, programmed death (apoptosis), and immune defense [53,54]. Oxidative stress triggered by ROS production can activate MAPK pathways, which can be monitored by assessing levels of phosphorylated ERK, JNK, and p38 MAPKs [18]. JNK1/2 and p38 MAPK pathways are usually activated by inflammatory cytokines and extracellular stressors such as UV light, heat, and ROS. Levels of phosphorylated JNK, p38, and ERK 1/2 MAPKs are increased in the postmortem brains of AD patients and inhibit their phosphorylation supply the methods to the treatment of AD [55,56]. Previous studies reported that many bioactive compounds isolated from MT could inhibit MAPK phosphorylation and exist neuroprotective effects [57,58]. Similarly, the current study revealed MT pretreatment notably prevented the H_2_O_2_-induced increases in levels of phosphorylated ERK, JNK, and p38 MAPKs. NF-κB is a protein complex that controls transcription of DNA, cytokine production, and cell survival and plays an important role in neurodegenerative diseases [59]. The activation of NF-κB is considered to be part of the stress response, and it can be activated by a variety of stimuli including oxidative stress. H_2_O_2_ can rapidly activate NF-κB, which participates in ROS-induced cell death [60]. MT effectively inhibited H_2_O_2_-induced phosphorylation of NF-κB. Thus, MT protected SH-SY5Y neuroblastoma cells from H_2_O_2_-evoked apoptosis by inhibiting MAPK and NF-κB signaling.

## 4. Materials and Methods

### 4.1. Chemicals

Polyphenolic compounds, 3-(4,5-dimethylthiazol-2-yl) -2,5-diphenyltetrazolium bromide (MTT), 2, 2′-azino-bis (3-ethylbenzthiazoline-6-sulfonic acid) disodium salt (ABTS), fluorometric intracellular ROS kit, penicillin/streptomycin, and 2-(4-amidinophenyl)-6-indolecarbami dine dihydrochloride (DAPI) were purchased from Sigma-Aldrich (St. Louis, MO, USA). Parishin derivatives were bought from Chengdu Biopurify Phytochemicals Ltd. (Chengdu, Sichuan, China). Fetal bovine serum (FBS) was obtained from Young in Frontier Company (Seoul, Korea). 2,2-Diphenyl-1-picrylhydrazyl (DPPH) was bought from MedChemExpress (Princeton, NJ, USA). Ham’s F-12 Nutrient Mix medium, BCA protein assay kit, and cDNA synthesis kit (ReverTra Ace qPCR RT Kit) were purchased from Thermo Fisher Scientific (Waltham, MA, USA). Eagle’s Minimum Essential Medium (EMEM) was obtained from ATCC (Manassas, VA, USA). H_2_O_2_ was purchased from Fujifilm Wako Pure Chemical (Osaka, Japan). The SYBR green qPCR Kit was supplied by TOYOBO (Osaka, Japan). Lactate dehydrogenase (LDH) cytotoxicity assay kit was bought from TAKARA (Shiga Prefecture, Japan). Clarity western chemiluminescent (ECL) substrate kit was supplied by Bio-Rad Laboratories (Hercules, CA, USA).

### 4.2. MT Fruit Collection and Extract Preparation

Fully ripe MT was purchased in early November 2019, from a farm located in Sunchang district (Jeonbuk, Korea). An MT specimen was stored at the Herbarium of the Department of Food Science and Technology, College of Agricultural Life Science, Jeonbuk National University, Korea.

Powdered MT fruit (100 g) was extracted in 70% aqueous ethanol mixture (500 mL) at room temperature for 20 min using a sonicator (Hwa Shin Instrument Co., Seoul, Korea) and centrifuged (4500× *g*, 10 min). The medicinal residue was re-extracted twice as described above. Supernatants from the three extractions were combined and vacuum evaporated to remove ethanol at 40 °C. The obtained MT extract was freeze-dried.

### 4.3. High-Performance Liquid Chromatography (HPLC) Analysis

HPLC analysis of polyphenolic compounds was performed using a Waters HPLC system (Milford, MA, USA) consisting of a 2690 separation module and 996 photodiode array detector (PDA) with a ZORBAX Eclipse XDB-C18 column (250 mm × 4.6 mm, 5 µm; Agilent Technologies, Inc., Santa Clara, CA, USA). The mobile phase consisted of solvent A (0.1% formic acid and 10% acetonitrile in deionized water) and solvent B (0.1% formic acid and 10% deionized water in acetonitrile). The ratio of the mobile phase was maintained at A:B = 100:0 (0–5 min), 83:17 (5–10 min), 80:20 (10–30 min), and 100:0 (30–35 min) at a flow rate of 1.0 mL/min. UV–VIS absorption spectra were recorded at 200–400 nm during HPLC analysis, and polyphenolic compounds were measured at 280 nm. To analyze parishin derivatives, the mobile phase consisted of 0.1% phosphoric acid in deionized water (solvent A) and 0.1% phosphoric acid in methanol (solvent B). The mobile phase was used at A:B = 85:15 (0–5 min), 45:55 (5–20 min), and 85:15 (20–35 min) at a flow rate of 0.8 mL/min. UV–VIS absorption spectra were recorded from 200–400 nm during HPLC analysis, and parishin derivatives were detected at 220 nm. Identification of compounds was based on comparisons of HPLC retention time and UV profiles with those of authentic standards. Quantification of individual compounds was based on calibration curves prepared from serial dilutions of stock solution (1000 µg/mL).

### 4.4. Radical Scavenging Properties of MT Extract

DPPH and ABTS assays are convenient and popular to apply the radical scavenging properties of compounds. The radical scavenging properties of MT fruit extract were evaluated as described previously [61].

DPPH and ABTS radical scavenging levels (%) were acquired using the following Formula (1):(1)$$\mathrm{Radical}\;\mathrm{scavenging}\;\mathrm{level}\;\left(\%\right)=\frac{A\;\mathrm{control}\;-A\;\mathrm{sample}}{A\;\mathrm{control}}\times 100$$

Radical scavenging properties of MT are presented as half-maximal inhibitory concentration (IC_50_) (µg/mL).

### 4.5. Cell Culture

Human dopaminergic neuroblastoma SH-SY5Y cells were obtained from Korean Cell Line Bank (KCLB), and cultured in EMEM/F12 (1:1) mixture supplemented with 10% FBS and antibiotics in a 5% CO_2_ humidified incubator at 37 °C. SH-SY5Y cells were sub-cultured or seeded into plates until approximately 90% confluence.

### 4.6. Cell Cytotoxicity Assay and Morphological Observations

SH-SY5Y neuroblastoma cells were seeded into 96-well plates at a density of 1 × 10^4^ cells/well. To evaluate the cytotoxicity of MT extract and 50% lethal concentration (LC_50_) of H_2_O_2_, MT (5–100 µg/mL) or H_2_O_2_ (50–500 µM) was treated with cells for 24 h. To induce oxidative stress, cells were pretreated with different concentrations of MT for 2 h and then co-cultured with freshly prepared H_2_O_2_ (LC_50_) for 24 h. Cell viability was assayed with the MTT method. In addition, images of cells were captured under an inverted microscope (400×) equipped with a camera (CKX41, Olympus Corporation, Shinjuku City, Japan) for morphological analysis.

### 4.7. DAPI Staining

Changes in the morphology of the nucleus of cells after H_2_O_2_ treatment were evaluated by 4, 6-diamidino 2-phenylindole dihydrochloride (DAPI) staining. SH-SY5Y cells were placed in a 24-well plate containing gelatin-coated coverslips. Cells from different treatment groups were processed and fixed with 4% paraformaldehyde (PFA) for 20 min at room temperature. Fixed cells were washed twice and permeabilized with 0.25% Triton X-100 in PBS for 10 min. Cells were incubated with fresh diluted DAPI (300 µM) solution for 5 min at room temperature.

Samples were rinsed with PBS and then water. Coverslips were removed from the wells carefully and mounted using mounting medium onto microscope slides. Stained cells were examined under a Leica DM2500 fluorescence microscope (Leica Microsystems, Wetzlar, Germany) at 400× magnification, and percentages of condensed nuclei and fragmented nuclei were calculated.

### 4.8. Intracellular ROS Production and Lactate Dehydrogenase (LDH) Release Assay

Intracellular ROS generation was detected using a fluorometric intracellular ROS Kit according to the manufacturer’s instructions. Briefly, SH-SY5Y cells were plated at a density of 1.0 × 10^4^ cells/well in 96-well plates overnight, pretreated with MT (5, 10, 20, and 50 mg/mL) for 2 h, and then co-treated with H_2_O_2_ for 3 h. Cells washed by Hanks’ Balanced Salt Solution (HBSS). The master reaction mix (100 µL) was added and then cells were incubated for 30 min in an incubator (5% CO_2_, 37 °C). The fluorescence intensity was measured at an excitation of 640 nm and an emission of 675 nm.

The release of LDH from cells was measured using an LDH cytotoxicity detection assay kit according to the manufacturer’s instructions. SH-SY5Y cells were plated at a density of 1.0 × 10^4^ cells/well in 96-well plates overnight, pretreated with MT (5, 10, 20, and 50 mg/mL) for 2 h, and then co-treated with H_2_O_2_ for 24 h. The supernatant was collected after 24 h, and LDH relative levels were detected by determining the absorbance at 490 nm with an automated microplate reader.

### 4.9. RNA Extraction and Quantitative Real-Time PCR (qPCR) Analysis

Total cellular RNA was isolated from cells with a commercial total RNA extraction kit (GeneAll, Seoul, Korea) according to the manufacturer’s instructions. cDNA was reverse-transcribed from 3 μg total RNA using the RevertAid First Strand cDNA Synthesis Kit according to the manufacturer’s instructions. Quantitative real-time PCR (qPCR) was performed on a CFX96 Real-Time PCR Detection System (Bio-Rad Laboratories, Hercules, CA, USA) with SYBR Green I as double-stranded DNA-specific binding dye. Specificity was verified by melting curve analysis. Quantification was performed by normalizing the *Ct* values of each sample to that of glyceraldehyde-3-phosphate dehydrogenase (GAPDH) as an internal control. All PCR primers were obtained from Bioneer (Daejeon, Korea) (Table 3).

### 4.10. Western Blot Analysis

SH-SY5Y cells were pretreated with MT (20 and 50 μg/mL) for 2 h and then co-incubated with H_2_O_2_ for 24 h. Cells were lysed with radioimmunoprecipitation assay (RIPA) buffer. The supernatant was collected after centrifugation at 13,000× *g* for 15 min at 4 °C. Equal amounts of proteins were loaded on 10–12% sodium dodecyl sulfate-polyacrylamide gels for electrophoresis (SDS-PAGE) and then transferred onto nitrocellulose membranes. Membranes were incubated in 5% non-fat dry milk in Tris-buffered saline with Tween-20 (TBST) at room temperature for 2 h. Primary antibodies were added, and membranes were incubated at 4 °C overnight after three washes. The primary antibodies listed in Table 4 were used according to the manufacturers’ specifications. Membranes were washed three times with TBST and then incubated with secondary antibodies (goat anti-rabbit IgG-HRP) for 2 h. Bands were visualized using a Clarity Western Chemiluminescent (ECL) substrate kit, and band images were obtained using the LAS 500 imaging system (GE Healthcare, Chicago, IL, USA). Bands were quantified using Quantity One software (Bio-Rad Laboratories, Hercules, CA, USA).

### 4.11. Statistical Analysis

All data are expressed as mean ± standard error of the mean (SEM). Group comparisons were performed using one-way analysis of variance (ANOVA) in Prism 7.0 (GraphPad Software Inc., San Diego, CA, USA). A value of *p* < 0.05 was considered statistically significant.

## 5. Conclusions

Taken together, our results demonstrate that MT can attenuate H_2_O_2_-induced neurotoxicity in SH-SY5Y cells. MT suppressed intracellular ROS generation and LDH leakage and increased the expression of genes encoding antioxidant enzymes as well as proteins involved in brain function and neurotransmitter synthesis. Furthermore, MT regulated the expression of apoptosis-related proteins (Bcl-2, Bax, and caspase 3), while pretreatment of cells with MT blocked the H_2_O_2_-induced phosphorylation of MAPK/NF-κB signaling pathway proteins. MT is a promising therapeutic agent for the treatment of neurodegenerative diseases.

## Figures and Tables

**Figure 1 ijms-22-06946-f001:**
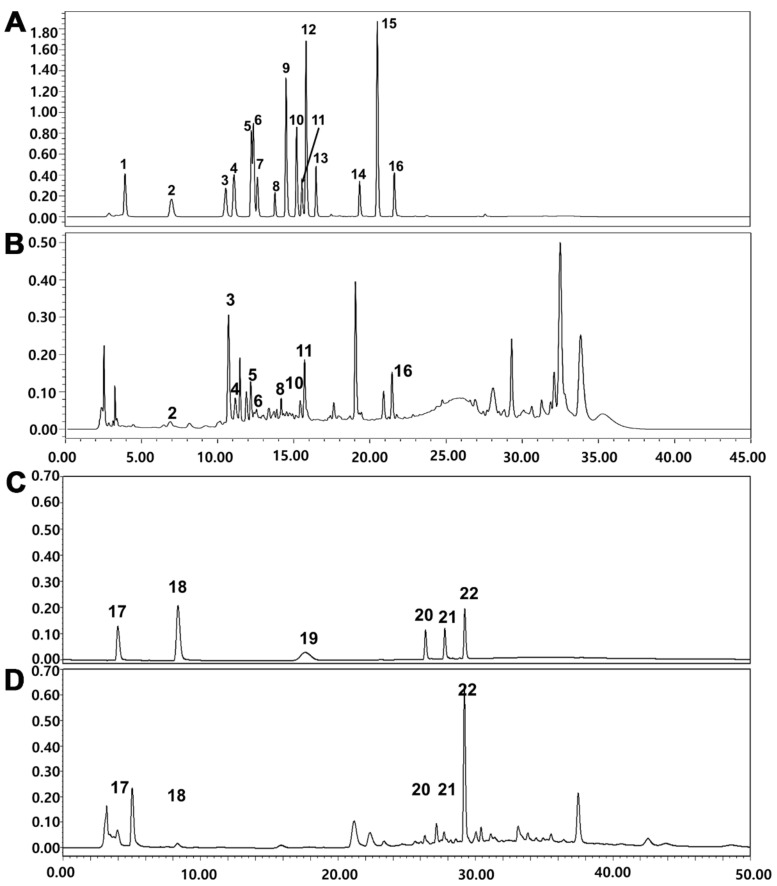
HPLC chromatograms of (**B**) polyphenolic compounds and (**D**) parishin derivatives of the 70% aqueous ethanol extract of MT. Mixtures of authentic standards for (**A**) polyphenolic compounds and (**C**) parishin derivatives: 1. gallic acid, 2. protocatechuic acid, 3. *p*-hydroxybenzoic acid, 4. chlorogenic acid, 5. caffeic acid, 6. syringic acid, 7. isovanillic acid, 8. rutin, 9. *p*-coumaric acid, 10. ferulic acid, 11. taxifolin, 12. trans-coumaric acid, 13. rosmarinic acid, 14. quercetin, 15. trans-cinnamic acid, 16. kaempferol, 17. gastrodin, 18. *p*-hydroxybenzyl alcohol, 19. parishin E, 20. parishin B, 21. parishin C, 22. parishin A.

**Figure 2 ijms-22-06946-f002:**
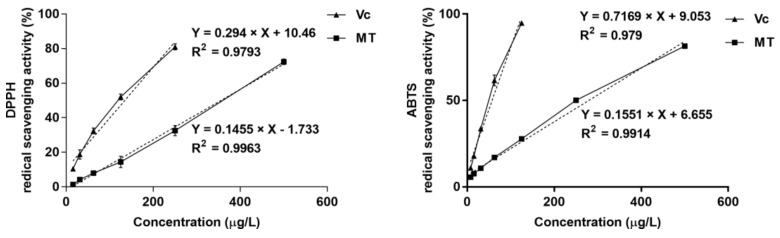
DPPH and ABTS radical scavenging activity of MT extract. MT exhibited antioxidant activity with an IC_50_ of 355.821 ± 8.343 µg/mL for DPPH and 278.741 ± 1.300 µg/mL for ABTS. Data are expressed as mean ± SEM, *n* =3.

**Figure 3 ijms-22-06946-f003:**
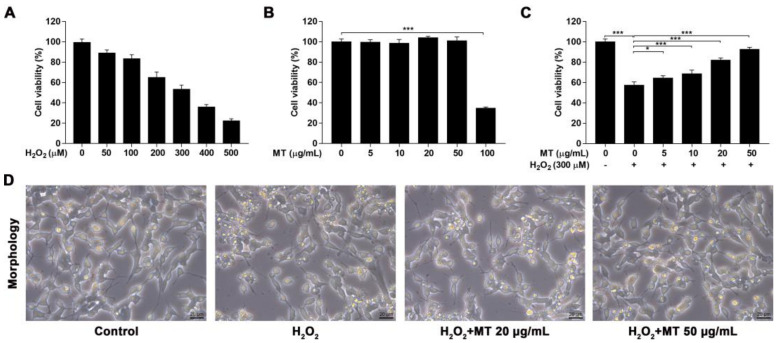
Protective effects of MT against H_2_O_2_-induced cell damage in SH-SY5Y cells. (**A**) SH-SY5Y cells were treated with various concentrations of H_2_O_2_ for 24 h, and LC_50_ (50% lethal concentration) was determined by MTT assay. (**B**) SH-SY5Y cells were treated with different concentrations of MT to assess cytotoxicity. (**C**) SH-SY5Y cells were pretreated with different concentrations of MT for 2 h before H_2_O_2_ treatment, and cell viability was measured. (**D**) Morphological changes (400×) were observed under an inverted microscope. Data are presented as mean ± SEM of three independent experiments. * *p* < 0.05, and *** *p* < 0.001.

**Figure 4 ijms-22-06946-f004:**
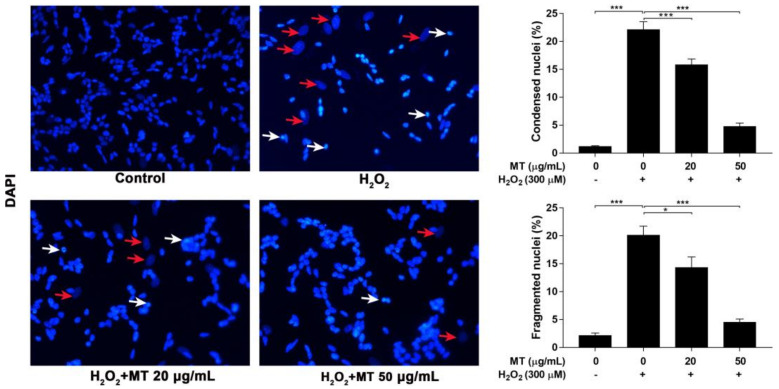
Effect of MT on H_2_O_2_-induced apoptosis in SH-SY5Y cells. Cells were pretreated with MT for 2 h following exposure to H_2_O_2_, percentages of condensed nuclei (white arrows) and fragmented nuclei (red arrows) were detected by DAPI staining (400×). Data are presented as mean ± SEM of three independent experiments. * *p* < 0.05, and *** *p* < 0.001.

**Figure 5 ijms-22-06946-f005:**
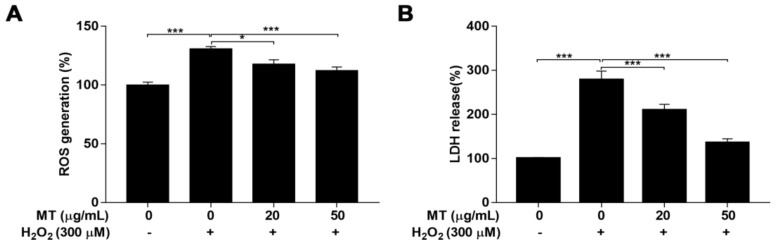
Effect of MT on H_2_O_2_-induced oxidative stress and cytotoxicity in SH-SY5Y cells. Cells were pretreated with MT for 2 h following exposure to H_2_O_2_. (**A**) ROS generation and (**B**) LDH release were assessed by ROS and LDH commercial kits according to the manufacturers’ instructions. Data are presented as mean ± SEM of three independent experiments. * *p* < 0.05, and *** *p* < 0.001.

**Figure 6 ijms-22-06946-f006:**
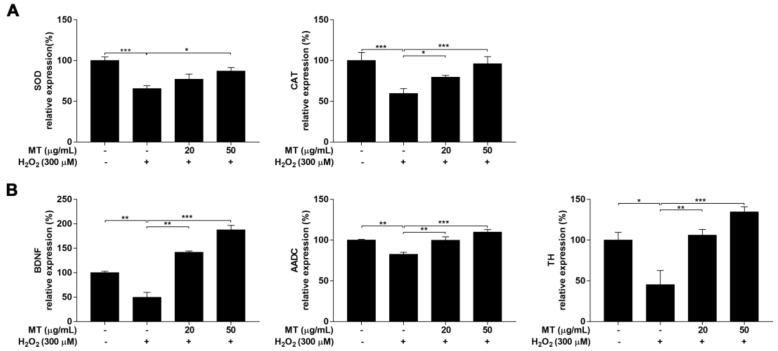
Effects of MT on the expression of genes encoding antioxidant enzymes and neuronal markers in SH-SY5Y neuroblastoma cells. Expression of genes encoding the antioxidant enzymes (**A**) SOD and CAT and the neuronal markers (**B**) BDNF, AADC, and TH were evaluated by real-time PCR. Data are presented as mean ± SEM of three independent experiments. * *p* < 0.05, ** *p* < 0.01, and *** *p* < 0.001.

**Figure 7 ijms-22-06946-f007:**
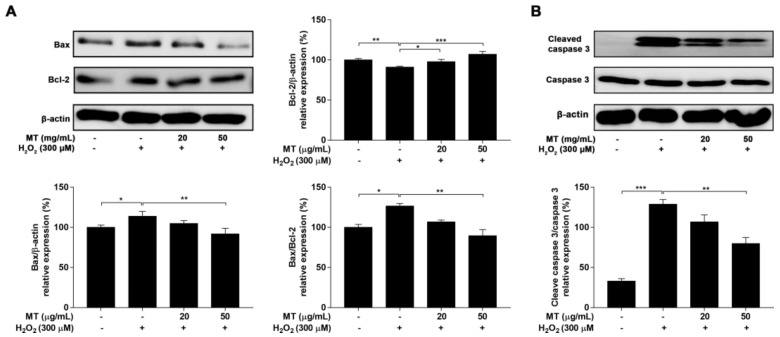
Effect of MT on the expression of apoptosis-related proteins in SH-SY5Y neuroblastoma cells. Expression levels of (**A**) Bcl-2, Bax, and (**B**) caspase-3 were analyzed by western blotting. Data are presented as mean ± SEM of three independent experiments. * *p* < 0.05, ** *p* < 0.01, and *** *p* < 0.001.

**Figure 8 ijms-22-06946-f008:**
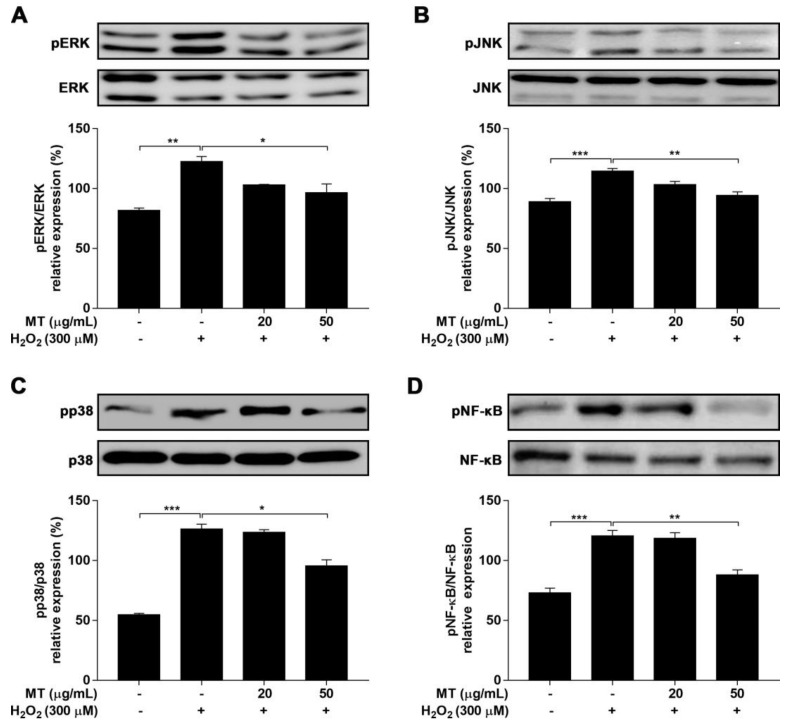
Effects of MT on MAPK (ERK, JNK, and p38) and NF-κB protein expression in SH-SY5Y neuroblastoma cells. Cells were pre-treated with MT for 2 h, followed by treatment with H_2_O_2_ for another 24 h. Expression levels of phosphorylated (**A**–**C**) MAPK family proteins and (**D**) NF-κB were evaluated by western blot analysis. Data are presented as mean ± SEM of three independent experiments. * *p* < 0.05 ** *p* < 0.01, and *** *p* < 0.001.

**Table 1 ijms-22-06946-t001:** Polyphenolic compounds and parishin derivatives identified in a 70% aqueous ethanol extract of MT fruit.

Compounds	Concentration (g/g dw)
Protocatechuic acid	69.4 ± 1.3
*p*-Hydroxybenzoic acid	1066.8 ± 2.7
Chlorogenic acid	354.4 ± 0.9
Caffeic acid	57.4 ± 0.4
Syringic acid	84.5 ± 0.6
Isovanillic acid	128.9 ± 1.1
Rutin	223.0 ± 1.6
Taxifolin	87.7 ± 0.8
Quercetin	283.6 ± 1.3
Kaempferol	534.0 ± 0.3
Gastrodin	1296.5 ± 2.5
*p*-Hydroxybenzyl alcohol	215.8 ± 1.2
Parishin B	656.5 ± 2.1
Parishin C	1015.2 ± 1.5
Parishin A	7021.4 ± 3.5

**Table 2 ijms-22-06946-t002:** Effects of MT and ascorbic acid on DPPH and ABTS free radical scavenging assays.

Plant Extract/Standard	IC_50_ Value (μg/mL) of Radical Scavenging	
	DPPH Radical	ABTS Radical
MT extract	355.821 ± 8.343	278.741 ± 1.300
Ascorbic acid	134.501 ± 0.555	57.123 ± 0.873

Note: All data were expressed as the mean ± SEM, *n* = 3.

**Table 3 ijms-22-06946-t003:** Sequences of primers using in this study.

Gene Target	Primer Name	Primer Sequence (5′-3′)
SOD	SOD F SOD R	AGGCCGTGTGCGTGCTGAAG CACCTTTGCCCAAGTCATCTGC
CAT	CAT F CAT R	CCTTTCTGTTGAAGATGCGGCG GGCGGTGAGTGTCAGGATAG
TH	TH F TH R	GAGGAGAAGGAGGGGAAG ACTCAAACACCTTCACAGCT
AADC	AAD F AAD R	AACAAAGTGAATGAAGCTCTTC GCTCTTTGATGTGTTCCCAG
BDNF	BDNF F BDNF R	ATGACCATCCTTTTCCTTACT GCCACCTTGTCCTCGGAT
GAPDH	GAPDH F GAPDH R	TTCACCACCATGGAGAAGGC GGCATGGACTGTGGTCATGA

Note: SOD, Superoxide dismutase; CAT, Catalase; TH, Tyrosine hydroxylase; AADC, Amino acid decarboxylase; BDNF, Brain-derived neurotrophic factor; GAPDH, Glyceraldehyde 3-phosphate dehydrogenase.

**Table 4 ijms-22-06946-t004:** Primary antibodies were used in this study.

Antibody	Host	Manufacturer	Cat. No.	Dilution
Bcl-2	Rabbit	Cell Signaling Technology, Danvers, MA, USA	3498	1:1000
Bax	Rabbit	Santa Cruz Biotechnology, Dallas, TX, USA	sc-493	1:500
Cleave caspase 3	Rabbit	Cell Signaling Technology, Danvers, MA, USA	9661	1:1000
Caspase 3	Rabbit	Cell Signaling Technology, Danvers, MA, USA	9665	1:1000
p-JNK	Rabbit	Cell Signaling Technology, Danvers, MA, USA	9251	1:1000
JNK	Rabbit	Cell Signaling Technology, Danvers, MA, USA	9252	1:1000
p-ERK	Rabbit	Cell Signaling Technology, Danvers, MA, USA	9101	1:1000
ERK	Rabbit	Cell Signaling Technology, Danvers, MA, USA	9102	1:1000
p-p38	Rabbit	Cell Signaling Technology, Danvers, MA, USA	9211	1:1000
p38	Rabbit	Cell Signaling Technology, Danvers, MA, USA	9212	1:1000
p-NF-κB	Rabbit	Cell Signaling Technology, Danvers, MA, USA	3033	1:1000
NF-κB	Rabbit	Cell Signaling Technology, Danvers, MA, USA	8242	1:1000
β-actin	Rabbit	Cell Signaling Technology, Danvers, MA, USA	4970	1:1000

## Data Availability

Not applicable.

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
