# Peer review of "Ethanol Extract of Maclura tricuspidata Fruit Protects SH-SY5Y Neuroblastoma Cells against H2O2-Induced Oxidative Damage via Inhibiting MAPK and NF-κB Signaling"

_ijms, 2021, doi:10.3390/ijms22136946_

Round 1

Reviewer 1 Report

The work about neuroprotector effect of extract Maclura tricuspidata against H2O2-induced oxidative damage via inhibiting MAPK/NF-κB signaling is an interesting work but I think that have some mistake.

The bibliographic citations are not numbers.

Introduction

I do not understand why only write about “suppression of nicotinamide adenine dinucleotide phosphate (NADPH) oxidase” if the authors after do not measure. Also, we do not write about mitochondria targeting of bcl/bax and the main source of ROS.

Result.

MTT is not methods for evaluating apoptosis,

Line 141. Analysis of DAPI is not the best method for apoptosis evaluation, see below.

Line 155. heading of Figure 3.6 “LDH Synthesis “is confusing I think is LDH release and Lactate Dehydrogenase (LDH) Production in 374 lines

Figure 8. The wb of p38 and NF-Kb and JNK are oversaturated and cause measurement inaccuracy.

ROS production: what kit was used? What type of ROS was measure?  What is the 52 citation?

Major

LINES:

 243-244. Analysis of DAPI is not the best method for apoptosis evaluation, perhaps Giemsa stained could be helping to evaluate cell and nuclear morphology.

184 and other- The expression of antioxidant enzymes must be proved with the level activity of those.

253-255. “Fluorescence intensity of cells is directly proportional to intracellular ROS accumulation promoted by H2O2, In this study, a rapid elevation in fluorescence intensity was observed in cells exposed to H2O2, indicating severe oxidative stress”.  Where has the author shown this result, which figures, method?

  1. “MT suppressed the generation of intracellular ROS and LDH”?. What is the significance of the generation of LDH? Perhaps it suppresses the cytotoxic activity of H2O2.

Author Response

The work about neuroprotector effect of extract Maclura tricuspidata against H2O2-induced oxidative damage via inhibiting MAPK/NF-κB signaling is an interesting work but I think that have some mistake.

The bibliographic citations are not numbers.

Introduction

Point 1: I do not understand why only write about “suppression of nicotinamide adenine dinucleotide phosphate (NADPH) oxidase” if the authors after do not measure. Also, we do not write about mitochondria targeting of bcl/bax and the main source of ROS.

Response 1: Thanks for your suggestions, and we already corrected and rewrote the sentences.

Result.

Point 2: MTT is not methods for evaluating apoptosis,

Response 2: Thank you for your comment, we already change the “apoptosis” to “cytotoxicity”.

Point 3: Line 155. Heading of Figure 3.6 “LDH Synthesis “is confusing I think is LDH release and Lactate Dehydrogenase (LDH) Production in 374 lines.

Response 3: Thank you very much, we have corrected.

Point 4: Figure 8. The wb of p38 and NF-Kb and JNK are oversaturated and cause measurement inaccuracy.

Response 4: Thank you for your valuable comments! We already found the original images, analyzed them again and changed the images.

Point 5: ROS production: what kit was used? What type of ROS was measure?  What is the 52 citation?

Response 5: Thank you for your comments! We have added the ROS information in detail.

Major

LINES:

Point 6: 243-244. Analysis of DAPI is not the best method for apoptosis evaluation, perhaps Giemsa stained could be helping to evaluate cell and nuclear morphology.

Response 6: Thank you for your valuable comments. Here DAPI staining was mainly used to evaluate the effects of MT in the nuclear morphological changes in H2O2-exposed SH-SY5Y cells. Previous research also shows that it can be used to observe the nuclear morphological changes, we already attached the reference.

Point 7: 184 and other- The expression of antioxidant enzymes must be proved with the level activity of those.

Response 7: Thank you very much! In this experiment, we performed qPCR (not ELISA) to analyze the antioxidant enzymes gene expression. We have corrected the expression sentences.

Point 8: 253-255. “Fluorescence intensity of cells is directly proportional to intracellular ROS accumulation promoted by H2O2, In this study, a rapid elevation in fluorescence intensity was observed in cells exposed to H2O2, indicating severe oxidative stress”.  Where has the author shown this result, which figures, method?

Response 8: Thank you very much! Fluorescence intensity is the value that we measure the intracellular ROS using a fluorescence plate reader. I am sorry to make you confuse, we already rewrote and corrected our expression.

Point 9: 417 “MT suppressed the generation of intracellular ROS and LDH”? What is the significance of the generation of LDH? Perhaps it suppresses the cytotoxic activity of H2O2.

Response 9: Thank you very much! We have corrected.

Reviewer 2 Report

This study investigated neuroprotective effects of Maclura tricuspidata extract (MT) against H2O2-induced oxidative damage in SH-SY5Y cells. MT improved cell viability, attenuated ROS production and increased gene expression of anti-oxidant enzymes, prevented apoptotic changes and activated MAPK and NF-κB signalling cascades. Although results are interesting there are some limitations that must be resolved before considering manuscript for acceptance.

Major

Figure 3D

Neuroprotective effect of 20 µg/ml MT is not visible from the photograph although viability should be around 80%.

Figure 4

In my opinion, the fluorescence density is not determined and presented correctly. It should be expressed per number of cells. Of course that fluorescence is lower when number of cells is dramatically reduced. Apoptotic cells have condensed nuclei (meaning higher fluorescence) and fluorescence, indicated per number of cells, should be higher, not lower. The other possibility is to count percentage of cells with nuclear condensation.

Again, at this resolution, the protective effect of 20 µg/ml MT is not evident.

Figure 7

Bcl-2 blot in Figure 7A is not convincing, all changes are between 5%, such a small changes cannot be detected by WB method, and error bars indicated are extremely small for this method

Bax expression, as well as Bax/Bcl-2 ratio for 50 µg/ml MT – blot and graph do not match

Ratio cleaved caspase-3/caspase 3 could not be over 50% for control group (cleaved caspase-3 is barely detectable)

Line 244 – LDH leakage happens when cell membranes are damaged which resembles more to necrotic death. It is more likely that SH-SY5Y cells died by both apoptosis and necrosis. Please reconsider interpretation of the obtained data.

Figure 8

If it is written MAPK/NF-κB signalling, this usually means that MAPK pathways activate NF-κB.  No evidence are provided that MAPK pathways are really involved in NF-κB activation. It can only be concluded that both of these signalling cascades are involved. If MAPK pathways are upstream of NF-κB, then selective inhibitors of ERK1/2, JNK and p38 should prevent activation of NF-κB pathway after exposure to hydrogen peroxide.

Figure 9 – NF-κB is missing, please add to the scheme

Discussion section largely repeats the results and should be rewritten. Obtained results should be compared with other studies using Maclura tricuspidata extract. Some studies also aimed to identify bioactive compounds of MT. Please compare results.

Throughout the manuscript, the authors often referred to Parkinson’s disease. Study performed on differentiated SH-SY5Y cells will be more relevant for Parkinson’s disease, as well as the use of more specific neurotoxin, such as MPP+ or 6-OHDA, that are also related to increased generation of ROS .

Minor issues

Line 19: oxidative stress-induced damage

Line 21: LDH is released from intracellular space into culturing media, it is not produced

Line 25 – specify what MAPK

Line 59 – cytochrome c generation or cytochrome c release?

Line 74 - oxidative-regulated damage – not grammatically correct

Table 1 – Kaempferol should be with one decimal point

Table 2 could be omitted as values are indicated in the text

Line 115 – should be Figure 2 instead of Figure 3

Figure 2 – please correct legend – there is no clear difference between the lines of MT and VC

Line 121 – To investigate protective effect of MT against ......

Line 128 – H2O2-induced cell death, at this step in the manuscript it is not known that cells died by apoptosis

Line 145 – all cells are DAPI-positive, this means nothing, please rewrite

Line 155 – change synthesis to leakage

On the legends of Figures 3-9 indicate 300 µM H2O2

Line 227 – what is the difference between free radicals and ROS?

Line 253 – in this study ROS were increased by 30%, increase is too small to indicate severe oxidative stress

Line 258 - the easy breach of antioxidant defense mechanism – unclear

Line 262 – decrease of ROS release – please explain

Line 272 - Caspase-3, which is activated by H2O2, plays a vital role in the apoptotic death of neurons [19]. – could play a vital role, neuronal death is often caspase-3 independent

Line 377 – please indicate name and the supplier of LDH assay

Author Response

This study investigated neuroprotective effects of Maclura tricuspidata extract (MT) against H2O2-induced oxidative damage in SH-SY5Y cells. MT improved cell viability, attenuated ROS production and increased gene expression of anti-oxidant enzymes, prevented apoptotic changes and activated MAPK and NF-κB signalling cascades. Although results are interesting there are some limitations that must be resolved before considering manuscript for acceptance.

Major

Point 1: Figure 3D

Neuroprotective effect of 20 µg/ml MT is not visible from the photograph although viability should be around 80%.

Response 1: Thank you very much! Each group we have taken several images, maybe this image is not representative, and we already check all the image and change it to a representative image.

Point 2: Figure 4

In my opinion, the fluorescence density is not determined and presented correctly. It should be expressed per number of cells. Of course that fluorescence is lower when number of cells is dramatically reduced. Apoptotic cells have condensed nuclei (meaning higher fluorescence) and fluorescence, indicated per number of cells, should be higher, not lower. The other possibility is to count percentage of cells with nuclear condensation.

Response 2: Thank you very much! The apoptotic process has several stages, from cellular shrinkage and chromatin condensation, concomitant with formation of membrane blebs. At the beginning of the apoptotic process, cells exhibited nuclear condensation staining (Figure 4, white arrow), and but at the middle stages of the apoptotic process, the nucleus begins to lyse, and cells exhibited nuclear sporadic weaker staining (Figure 4, red arrow). In our experiment, both of stages can be observed. At the end stage of the apoptotic process, organelles and nucleus fragment and the blebs begin formation of apoptotic bodies which are eventually engulfed by macrophages or neighboring cells by endocytosis/phagocytosis, and showed DAPI-negative staining. Apoptotic cells have condensed nuclei, but the size of the nuclei is smaller than normal cells. The total fluorescence of condensed nuclei and normal nuclei is almost similar. We thought that fluorescence density of all cell can reflect the condition of the cell. I hope my answer will satisfy you. Thank you very much again!

Point 3: Again, at this resolution, the protective effect of 20 µg/ml MT is not evident.

Response 3: Thank you for comment! Each group we have taken several images, maybe this image is not representative, and we already check all the image and change to a representative image.

Point 4: Figure 7

Bcl-2 blot in Figure 7A is not convincing, all changes are between 5%, such a small changes cannot be detected by WB method, and error bars indicated are extremely small for this method

Bax expression, as well as Bax/Bcl-2 ratio for 50 µg/ml MT – blot and graph do not match

Response 4: Thank you very much. We have checked the raw data, and all changes are more than 10%. Moreover, all data are expressed as mean ± standard error of the mean (SEM), so the error bars are much smaller than the data are expressed as mean ± standard deviation (SD).

I am sorry for that, and we have measured the bands again and corrected the Figure 7.

Point 5: Ratio cleaved caspase-3/caspase 3 could not be over 50% for control group (cleaved caspase-3 is barely detectable)

Response 5: Thank you very much. We have checked our raw data, and found the differences in cleaved caspase-3 protein expression between groups. The cleaved caspase-3 has two bands (17 and 19kDa), and its distribution area is larger than that of caspase 3, so the bands look a little fuzzy. We put the two images together and then use the Quantity One – 4.6.6 software to measure, the relative expression are 37.96% (cleaved caspase-3) and 62.04% (caspase-3), and the ratio is 61.12%, more than 50%. Moreover, we checked the other researches related to the ratio of cleaved caspase-3/caspase 3 and found that our result is similar to theirs or even some famous company. We have attached the link.

https://www.mdpi.com/1420-3049/22/10/1646

https://www.ncbi.nlm.nih.gov/pmc/articles/PMC5346704/

https://www.abcam.com/apoptosis-western-blot-cocktail-prop17-caspase-3-cleaved-parp1-muscle-actin-ab136812.html

I hope my answer will satisfy you, thank you again for your valuable comments

Point 6: Line 244 – LDH leakage happens when cell membranes are damaged which resembles more to necrotic death. It is more likely that SH-SY5Y cells died by both apoptosis and necrosis. Please reconsider interpretation of the obtained data.

Response 6: Thank you very much. We already corrected the explanation.

Point 7: Figure 8

If it is written MAPK/NF-κB signalling, this usually means that MAPK pathways activate NF-κB.  No evidence are provided that MAPK pathways are really involved in NF-κB activation. It can only be concluded that both of these signalling cascades are involved. If MAPK pathways are upstream of NF-κB, then selective inhibitors of ERK1/2, JNK and p38 should prevent activation of NF-κB pathway after exposure to hydrogen peroxide.

Response 7: Thank you very much. We already corrected the expression and change it to “MAPK and NF-κB signaling pathways.

Point 8: Figure 9 – NF-κB is missing, please add to the scheme

Response 8: Thank you very much! We already added it in the Figure and make the figure as the graphical abstract.

Point 9: Discussion section largely repeats the results and should be rewritten. Obtained results should be compared with other studies using Maclura tricuspidata extract. Some studies also aimed to identify bioactive compounds of MT. Please compare results.

Response 9: Thank you very much! We already rewrote the discussion section.

Point 10: Throughout the manuscript, the authors often referred to Parkinson’s disease. Study performed on differentiated SH-SY5Y cells will be more relevant for Parkinson’s disease, as well as the use of more specific neurotoxin, such as MPP+ or 6-OHDA, that are also related to increased generation of ROS.

Response 10: Thank you very much. Superoxide and hydrogen peroxide (H2O2) are the main ROS formed by mitochondria. We would like to investigate the protective effect in specific neurotoxin-induced the neurotoxicity, such as MPP+ or 6-OHDA.

Point 11: Minor issues

Line 19: oxidative stress-induced damage

Line 21: LDH is released from intracellular space into culturing media, it is not produced

Line 25 – specify what MAPK

Line 59 – cytochrome c generation or cytochrome c release?

Line 74 - oxidative-regulated damage – not grammatically correct

Table 1 – Kaempferol should be with one decimal point

Table 2 could be omitted as values are indicated in the text

Line 115 – should be Figure 2 instead of Figure 3

Figure 2 – please correct legend – there is no clear difference between the lines of MT and VC

Line 121 – To investigate protective effect of MT against ......

Line 128 – H2O2-induced cell death, at this step in the manuscript it is not known that cells died by apoptosis

Line 145 – all cells are DAPI-positive, this means nothing, please rewrite

Line 155 – change synthesis to leakage

On the legends of Figures 3-9 indicate 300 µM H2O2

Line 227 – what is the difference between free radicals and ROS?

Line 253 – in this study ROS were increased by 30%, increase is too small to indicate severe oxidative stress

Line 258 - the easy breach of antioxidant defense mechanism – unclear

Line 262 – decrease of ROS release – please explain

Line 272 - Caspase-3, which is activated by H2O2, plays a vital role in the apoptotic death of neurons [19]. – could play a vital role, neuronal death is often caspase-3 independent

Line 377 – please indicate name and the supplier of LDH assay

Response 11: Thank you very much, we already corrected the minor issues.

Round 2

Reviewer 1 Report

I think that the response it is ok

Author Response

Point 1: I think that the response it is ok

Response 1: We are very grateful for the comments of the reviewers. Thank you very much!

Reviewer 2 Report

DAPI staining

I cannot agree with the explanation. In my experience, following exposure to hydrogen peroxide, fluorescence should be increased in the case of apoptotic events. Figure 4 - looking at images (photographs),  in H2O2-treated cells fluorescence is increased (as expected), although decrease is represented in graphical representation, probably because of the lack of data normalization (number of cells must be considered). Photographs with MT are not so clear (they are relatively bright), it seems that these photographs were taken at higher magnification.

Red and white arrows should be explained in the figure legend.

https://www.ncbi.nlm.nih.gov/pmc/articles/PMC5984558/

https://www.spandidos-publications.com/10.3892/or.2012.2170#

Bax and Bcl-2

Changes of Bcl-2 and Bax up to 10%  are not typical for classical apoptosis. I suggest to comment (in the Discussion section – lines 269-279) very faint changes that were observed. For example, in the paper that was cited in response letter (round 1), changes of Bax and Bcl-2 expression were up to 300%, not up to 10%.

https://www.mdpi.com/1420-3049/22/10/1646/htm

Cleaved caspase-3/caspase-3 ratio

It is without doubt that caspase-3 is highly activated, but mathematically, the results presented are not correct. Looking at blots, it is not possible that ratio of cleaved caspase-3 and caspase 3 is over 50% in control cells. Ratio of 50% means that bend intensity of caspase 3 is twice as the bend intensity of cleaved caspase-3. In control SH-SY5Y, cleaved caspase 3 is barely detectable (which is expected). If the ratio cleaved caspase-3/caspase 3 is 60%, this means that control cells have 60% of caspase-3 in cleaved form (which is unexpected for control cells and is not visible on blots). Something that is confusing to me, will also be confusing for the readers.

https://journals.plos.org/plosone/article?id=10.1371/journal.pone.0180953 (Fig. 10)

Line 234 – neural cells do not proliferate

Line 247 - previous

Author Response

Comments and Suggestions for Authors

Point 1: DAPI staining

I cannot agree with the explanation. In my experience, following exposure to hydrogen peroxide, fluorescence should be increased in the case of apoptotic events. Figure 4 - looking at images (photographs), in H2O2-treated cells fluorescence is increased (as expected), although decrease is represented in graphical representation, probably because of the lack of data normalization (number of cells must be considered). Photographs with MT are not so clear (they are relatively bright), it seems that these photographs were taken at higher magnification.

Red and white arrows should be explained in the figure legend.

https://www.ncbi.nlm.nih.gov/pmc/articles/PMC5984558/

https://www.spandidos-publications.com/10.3892/or.2012.2170#

Response 1: Thank you very much! We agree with you and have changed my analysis method and expressed it as the percentages of cells with condensed and fragmented nuclei. (https://www.nature.com/articles/s41598-018-30652-x). Cell exposure to hydrogen peroxide increased the condensed nuclei, which means increased fluorescence. At the same time, we have checked the original image and corrected the image

We already explained the red and white arrows in the figure legend.

Point 2: Bax and Bcl-2

Changes of Bcl-2 and Bax up to 10% are not typical for classical apoptosis. I suggest to comment (in the Discussion section – lines 269-279) very faint changes that were observed. For example, in the paper that was cited in response letter (round 1), changes of Bax and Bcl-2 expression were up to 300%, not up to 10%.

https://www.mdpi.com/1420-3049/22/10/1646/htm

Response 2: Thank you for your suggestions and comments! We have corrected our expression in the discussion part.

 Point 3: Cleaved caspase-3/caspase-3 ratio

It is without doubt that caspase-3 is highly activated, but mathematically, the results presented are not correct. Looking at blots, it is not possible that ratio of cleaved caspase-3 and caspase 3 is over 50% in control cells. Ratio of 50% means that bend intensity of caspase 3 is twice as the bend intensity of cleaved caspase-3. In control SH-SY5Y, cleaved caspase 3 is barely detectable (which is expected). If the ratio cleaved caspase-3/caspase 3 is 60%, this means that control cells have 60% of caspase-3 in cleaved form (which is unexpected for control cells and is not visible on blots). Something that is confusing to me, will also be confusing for the readers.

https://journals.plos.org/plosone/article?id=10.1371/journal.pone.0180953 (Fig. 10) 

Response 3: Thank you very much! We agree with you and sorry to misunderstand your meaning last time. Previously we just compare the relative protein expression between groups but not compare the relative protein expression to their own baseline. And the data showed only the relative protein expression between groups but not showed the relative protein expression to their own baseline. Now we carefully checked the raw data, measured the bands intensity, and analyzed again including relative protein expression to their own baseline. The average ratio of cleaved caspase 3/caspase 3 is 33.06%, which matched the band. Thanks!

Point 4: Line 234 – neural cells do not proliferate

Line 247 – previous

Response 4: Thank you very much! We have corrected the minor issues.

Round 3

Reviewer 2 Report

can be accepted